# Factors Affecting Traditional Medicinal Plant Knowledge of the Waorani, Ecuador

**Holger Weckmüller [1],\*, Carles Barriocanal [1,2], Roser Maneja [1] and Martí Boada [1]**

[1]   Institut de Ciència i Tecnologia Ambientals, Universitat Autònoma de Barcelona,
      Edifici C Campus de la UAB, 08193 Cerdanyola del Vallès, Catalonia, Spain
[2]   Departament de Geografia, Universitat de Barcelona, Montalegre 6, 08001 Barcelona, Catalonia, Spain
\*   Correspondence: holgerweckmueller@gmail.com

**Abstract:** This paper explores how medicinal plant knowledge of the Waorani (Ecuador) varies with socio-economic and demographic factors. Medicinal plant knowledge was compared at individual and community levels. Semi-structured interviews were performed with 56 informants (men N = 29, women N = 27) between 15 and 70 years old in five Waorani communities located within the Yasuní National Park and Waorani Ethnic Reserve. We found a positive correlation between an informant's medicinal plant knowledge and age, and a negative correlation between informant's medicinal plant knowledge and the years of schooling. Reasons behind these findings are thought to be in the rapid socio-cultural changes of the Waorani due to globalization processes. Increased accessibility to health centers and improved transportation infrastructure result in a loss of ethnobotanical knowledge.

**Keywords:** ethnobotany; traditional ecological knowledge; indigenous communities; Ecuadorian amazon; loss of knowledge; globalization; global change; acculturation; socio-cultural changes

---

## 1. Introduction

Paradoxically, academic interest on ethnobotany, and specifically medicinal plants, is increasing while rural and indigenous people's knowledge about the use of plants for medicinal purposes is declining [1]. Studies around the world have shown that elders, in general, tend to know more about medicinal plants than younger generations [2–11]. Voeks and Leony [9] explain the phenomena of finding a greater knowledge within older generations in that people acquire more knowledge with age. Nevertheless, other authors claim that the majority of acquisition of traditional skills, including knowledge on plants, happens before the age of 15 [11–14]. However, the differences in knowledge might not be explained by the greater life experience of the elders, but by other socio-economic factors.

Studies show that, in general, women know more than men about medicinal plants [7,9,15]; some authors suggest that this is because men are more exposed at losing ethnobotanical plant knowledge than women [16]. Women's work activities in home gardens and as primary family health caregivers might explain this finding [17,18]. On the other hand, in some indigenous societies' men are more likely to leave their communities in order to find new economic activities [16]. Consequently, Hanazaki et al. [19] discovered that in some communities in the Atlantic rainforest of Brazil, men know more medicinal plants than women, which might mean that men in those communities have a closer relationship with the forest.

In addition, westernized schooling has been negatively associated with persistence of traditional ecological knowledge [9,20,21]. Traditional ecological knowledge (TEK) is defined as the body of knowledge, beliefs, traditions, practices, institutions, and worldviews locally developed and sustained by indigenous and rural communities in interaction with their physical environment [22]. Even if the curriculum promotes conceptual learning of TEK, the absence of direct contact with nature can change

the traditional transmission and acquisition ways [23]. Beyond such individual conditions, the many facets of modernization of rural societies may lead to a loss of traditional ecological knowledge [20,24,25]. Community location and infrastructure, such as proximity to urban areas, accessibility, and the presence of public healthcare, may account for medicinal plant knowledge loss [10,26,27]. For instance, in Manus Island, Papua New Guinea, the presence of public health systems contributed to local acculturation and the loss of knowledge about useful plants [28]. Vandebroek et al. [28] found that people in Isiboro-Sécure National Park have a greater knowledge of medicinal plants when they live in the communities that are furthest from the closest village and primary health care service.

Modernization of rural and indigenous societies is also influencing the Waorani, an indigenous group who were originally hunter-gatherers with a semi-nomadic lifestyle and now are settled in communities and what is now the Yasuní Biosphere Reserve in Ecuador, which includes the Yasuní National Park and the Waorani Ethnic Reserve. There they have become embedded in a strong modernization process during the last 50 years, which may have affected their knowledge about medicinal plants. In the Yasuní National Park, a road and a free bus system allow Waorani community members to get to urbanized areas in a short amount of time. Free health care is available at the medical center of a closely located oil company. Further, inhabitants have the opportunity to attend public school. Given the potential impacts of modernization on the knowledge of the Waorani's traditional livelihood, many studies have focused on collecting a range of knowledge about plant use in the Waorani society [29–32]. Little emphasis, however, has been on how such knowledge is being affected by the continuous changes of the socio-cultural environment in Waorani society. To better understand these dynamics and determining factors that play a role in the maintenance and loss of ethnobotanical knowledge, a cross-case analysis was conducted with three communities in the Yasuní National Park, which have access to health care and markets, and two communities outside the national park, at the Waorani Ethnic Reserve, which lack such access. This paper explores how Waorani medicinal plant knowledge varies in accordance with both socio-economic and demographic factors, using indicators of cultural change and modernization. Medicinal plant knowledge is compared not only at the individual level but also between different communities. To have a meaningful measure and to avoid bias, the theoretical dimension of knowledge about plants is combined with measures of the practical dimensions [26]. Further, this paper makes use of an easy methodology to capture viable information about traditional knowledge in a relatively short amount of time and without the substantial economic expense.

*The Waorani Livelihoods*

The Waorani are an indigenous society of hunters, gatherers, and horticulturists living in the eastern part of Ecuador, in the provinces of Napo, Orellana, and Pastaza. Unofficial estimates count the Waorani population at around 3800 individuals, which make up about 47 communities in an area over Waorani Ethnic Reserve and Yasuní National Park that is limited by the river Tiputini in the North, the river Curraray in the South, the foothills of the Andes, and the border of Peru [33,34]. Until missionaries made the first peaceful contact in 1958, the Waorani lived isolated as hunter-gatherer groups and led a semi-nomadic lifestyle in the lowlands of the Amazon forest [35]. The contact with missionaries brought them closer to contemporary society, introducing them to formal education, suspending activities of traditional medicine and shamanism, and establishing modern medicinal practices such as vaccination [33,35–37]. Oil exploitation in the territory had further repercussions in the socio-cultural dynamics of the Waorani [33]. The co-existence with oil companies and their workers brought western culture into the Waorani. However, not every Waorani community lives under the same conditions. While some communities neighbor oil companies and have easy access to health centers or market towns via the roads the companies built, others are only accessible by foot, boat, or plane. Some communities maintain self-sufficiency, while others have developed a paternalistic dependency on the oil companies in their territory or have been involved in the tourism industry. These differences in livelihood conditions make the Waorani communities an interesting case to research as

to why cultural goods, such as ethnobotanical knowledge, are declining in some cases while being maintained in others.

## 2. Materials and Methods

The study was conducted from April to June 2015 in 5 Waorani communities located inside the area of the Yasuní National Park and the Waorani Ethnic Reserve (Figure 1).

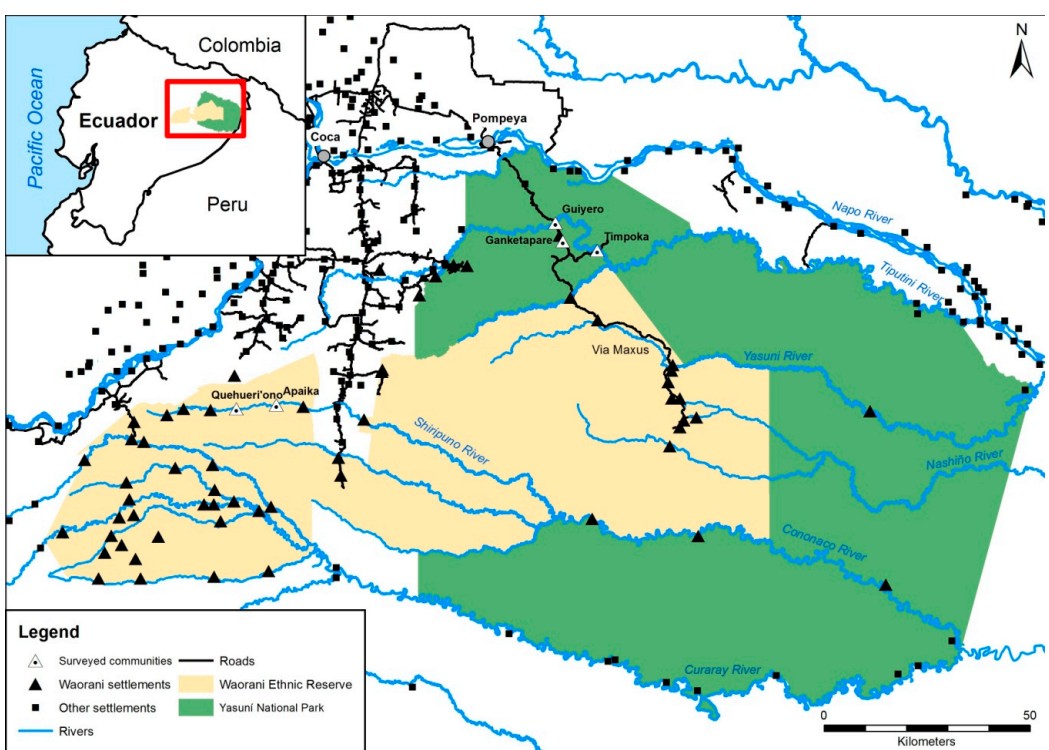

**Figure 1.** Study area with the location of the Waorani communities.

For comparative purposes, fieldwork was conducted in 3 communities located within the Yasuní National Park (Guiyero, Ganketapare, Timpoka) and 2 communities inside the Waorani Ethnic Reserve (Quehueri'ono and Apaika). Communities within the park had easy access to free health care at the medical center of a closely located oil company by the road via Maxus and a free bus system provided by this company. Further down the road is the Napo River where the Waorani can leave Yasuní National Park via ferry and access the closest small market town of Pompeya. Timpoka is the community nearest to the health center and Guiyero is the most distant. Furthermore, Guiyero has a school and is closer to the market town than Ganketapare and Timpoka.

Quehueri'ono and Apaika are located 35 and 50 km away from the Yasuní National Park, next to the Shiripuno River. Differently from those communities situated inside the Park, Quehueri'ono and Apaika are not connected with any roads, nor do they have easy access to a health center. The communities were accessible by foot, boat, or in the case of Quehueri'ono, by airplane. The closest health center was located in the city of Coca, a 4 hour motorboat and 2 hour bus ride far from Apaika and 2 hours further by motorboat from Quehueri'ono. Quehueri'ono has a landing strip for small airplanes which is used to bring food, school material to the village or tourists to the close situated *Waorani Ecolodge*. Quehueri'ono has a school whereas Apaika does not (Table 1).

**Table 1.** Characteristics of the five selected Waorani's communities chosen for the study.

| Community | Households | Households Participated | Informants | Access | | | Travel Time to | | |
|---|---|---|---|---|---|---|---|---|---|
| | | | | River | Aviation | Road | Health Center | School | Closest Market |
| Guiyero | 12 | 10 | 17 | yes | no | yes | <0.5 h by bus | 0 h | <1 h by bus |
| Ganketapare | 2 | 2 | 5 | no | no | yes | <0.5 h by bus | <0.25 h by bus | 1 h by bus |
| Timpoka | 7 | 6 | 16 | yes | no | yes | <0.5 h by bus | 0.5 h by bus | 1.5 h by bus |
| Quehueiri'ono | 12 | 10 | 16 | yes | yes | no | 6 h by motorboat + 2 h by bus | 0 h | 6 h by motorboat |
| Apaika | 1 | 1 | 2 | yes | no | no | 6h by motorboat + 2 h by bus | 2 h by boat | 4 h by motorboat |



### 2.1. Data Collection

Semi-structured interviews were used to gather information on ethnobotanical knowledge about medicinal plants and socio-economic characteristics. To design the ethnobotanical knowledge questions and select the plants, a previous bibliographic review was done at the library of the Pontificia Universidad Católica de Ecuador (PUCE). Every book or paper related to both topics, plants, and Waorani, was selected. Correspondingly 8 documents were reviewed [29–32,37–40]. The number of times each plant appeared in the sources was counted in order to find out the most studied plants in the literature, which were assumed to be the best known among the Waorani. With the help of academic experts in Waorani ethnobotany, 10 medicinal plants with their main uses were chosen. The criteria were that the chosen plant had to be a popular species that was easy to find. Every plant had at least one medicinal use and a proper way to be prepared, which was the second selection criterion (Table 2). Two of the plants appeared among the most cited plants of the bibliographic review. The others were less cited but fulfilled the latter criterion.

**Table 2.** Selected plants with respective uses and preparations.

| Species | Family | Growth Form | Part Used | Wild/Cultivated | Use | Preparation |
|---|---|---|---|---|---|---|
| *Abuta grandifolia* (Mart.) Sandwith | Menispermaceae | Shrub | steem, root | wild | Stomacache, diarrhea, stomach parasites | Scrached bark or root boiled in water. Drink the liquid. |
| *Croton lechleri* Müll. Arg. | Euphorbiaceae | Tree | sap | wild | Epidermic infections | Apply the sap of the tree directly to the skin. |
| *Eucharis x grandiflora* Planch. & Linden | Amaryllidaceae | Herb | bulb | wild | Abcess | Apply smashed bulb directly on the abscess. |
| *Euterpe precatoria* Mart. | Arecaceae | Tree | roots | wild | Influenza | The young red roots are boiled in water. Drink the liquid. |
| *Fittonia albivenis* (Lindl. ex Veitch) Brummitt | Acanthaceae | Herb | fruits | wild | Pimples | Apply smashed fruits directly on the skin. |
| *Iryanthera paraensis* Huber | Myristicaceae | Tree | sap | wild | Fungus | Apply sap directly on the affected parts. |
| *Musa x paradisiaca* L. | Musaceae | Herb | sap | cultivated | Diarrhea | Drink sap of the trunk. |
| *Neea* sp. | Nyctaginaceae | Tree | fruits | wild | Prevent cavities | Chew fruits. |
| *Theobroma subincanum* Mart. | Sterculiaceae | Tree | bark | wild | Fever | Cook macerated bark in water. Drink the liquid. |
| *Uncaria guianensis* (Aubl.) J.F.Gmel. | Rubiaceae | Vine | bark | wild | Influenza and cough | Cook macerated bark in water. Drink the liquid. |

At least one adult per household in each community was interviewed, except in households that did not agree to participate. The informant had to fulfill the following criteria: (1) Having a Waorani mother and/or father, (2) being fluent in Wao-tededo, the language of the Waorani. (3) Being older than 15 years old since at this age the majority of traditional ecological knowledge acquisition has already occurred [11–14], (4) not having behavioral or mental disorders, and (5) having lived at least 2 years in the community.

Interviews lasted between 30 min and 2 h and were conducted in Spanish. If the informant did not understand Spanish, the interview was conducted in Wao-tededo with the help of a local translator.

During the 56 interviews (men n = 29, women n = 27), pictures of each selected chosen plant, names, and identification of each verified by the botanical experts staff of Yasuní Scientific Station, were shown to the interviewee and s/he was asked if s/he recognized the plant, what Waorani people used it for, and how it was prepared. In addition, the following socio-economic questions were asked: (1) Years in educational systems (school and university); (2) years living in a city; (3) years working for a company; (4) hours of internet per week; (5) age, and (6) gender.

### 2.2. Data Analysis

A score was generated for the ethnobotanical knowledge about medicinal plants of each individual by adding (a) one point if s/he recognized the main use of the plant identified in the academic literature;

(b) one point if s/he mentioned the correct preparation for that use. In doing that, ethnobotanical knowledge score for each plant varied from zero (if use and preparation were not correct) to 2 (if both use and preparation were known). Therefore, with the 10 plants, the maximum ethnobotanical knowledge score possible would be 20, while the minimum would be 0.

Regarding the socio-economic variables, if the interviewee attended school or lived less than 3 months in a city at the moment of data analysis, the time was considered zero. A person working less than 3 months for a company or using the internet less than 30 min per week was also scored 0. Years living in a city, years working for a company, and hours of internet per week, due to weak numbers, were not considered in later data analysis. The Wao name of the plant was not considered [29–31].

To analyze the gathered data, the statistical program R was used. A linear regression model was used to understand relations between ethnobotanical knowledge, age, and time in schooling systems. Differences in the knowledge between gender and different types of communities were determined by Welch two sample t-tests. In addition, to find if potential differences between those informants with both Waorani parents and those with only one Waorani parent, a Welch two sample *t*-test was used.

## 3. Results and Discussion

### 3.1. Medicinal Plant Knowledge Among the Waorani

The range of medicinal plant knowledge was between 0 and 16 points and the average was 8.03 points. The maximum value that each person could reach was 20 points. All plant uses and preparations were recognized at least once, only the preparation of *Theobroma subincanum* was not recognized by anyone. Considering the sum of all interviews, 40% of the questions about medicinal plants use and preparation were answered (Table 3). In the more isolated communities, the mean of medicinal plant knowledge was 9.03 points, while lower in the more market-integrated ones (6.8 points).

**Table 3.** Frequencies of answers about the use and preparation of selected plants.

| Variable | Minimum | Maximum | Absolute Frequency | Relative Frequency |
|---|---|---|---|---|
| Total of communities | 0 | 1180 | 460 | 0.39 |
| Interviewees with both Waorani parents | 0 | 880 | 358 | 0.406 |
| Interviewees with one parent Waorani | 0 | 240 | 92 | 0.383 |

### 3.2. Gender and Knowledge

No significant relation was discovered between gender and medicinal plant knowledge (t = 0.7569, df = 53,169; *p* = 0.4525). This finding was inconsistent with many other studies, which indicated that generally in rural societies, women had greater ethnobotanical knowledge than men, at least considering medicinal plants [7,9,15]. The cases of higher medicinal plant knowledge among women might be explained by their role as primary family health caregivers [17,18]. The equal knowledge between Waorani men and women might be caused by the close connection of the community members with the forest. Even if hunting was a man's domain, it was usual to see women accompany their husbands during this activity. Besides, women habitually stay in the forest for gathering activities. Among the Waorani, healthcare was not the responsibility of only a few persons or of the housewives, rather almost all interviewees, independent of their age, claimed that during the last time they used traditional medicine it was themselves who gathered the plant and prepared the medicine.

### 3.3. Cross-Cultural Marriages and Knowledge

The analysis shows that there was no significant difference in the medicinal plant knowledge between people with one or both Waorani parents (t = 0,56896, df = 35,326; *p* = 0.573), which was on average 8.13 points in people with both parents and 7.66 in people with only one parent Waorani. A study about the botanical knowledge in Manus Island, Papua New Guinea had different results.

Cross-cultural marriages caused a reduction in botanical knowledge [27]. The authors conjecture that this is explained by reduced cultural pride due to the intermarriages between groups. Waorani communities do not show this trend because all participants, even if they only have one Waorani parent, grew up in Waorani communities surrounded by Waorani culture. Further, as these communities have not been isolated from deep socio-cultural changes, another way of understanding this result might be that cross-cultural marriage does not have any relevant effects in ethnobotanical knowledge, when it was already low, as the low general average score of the studied Waorani communities indicated.

### 3.4. Age and Knowledge

As expected, interviewees' medicinal plant knowledge and age had a positive significant correlation ($R^2 = 0.43$, $p < 0.001$) (Figure 2). Thus, elders tended to know more about medicinal plants than their younger counterparts. Several other authors found the same tendency in their studies about medicinal plant knowledge in different parts of the world [2,3,5–10,27,41]. Reasons for greater knowledge with increasing age could be numerous and have to be interpreted with caution. It might be logical that with progressive age, people have more time to accumulate knowledge and, therefore, show greater medicinal plant knowledge than the younger generation [9]. However, the correlation between knowledge and age does not necessarily mean an increase of ethnobotanical knowledge over time. Other authors see the reason for lesser knowledge in the younger population in ongoing socio-economic and cultural changes. Figueireido et al. [15] claim that younger people in an Atlantic rain forest community in Brazil are less interested in homemade medicine and more drawn to modern medicine. In rural communities of Cabo Delgado, Mozambique, such knowledge tends to be lost between generations because the younger people are more receptive to modern health centers than to the medicinal knowledge of their elders [2].

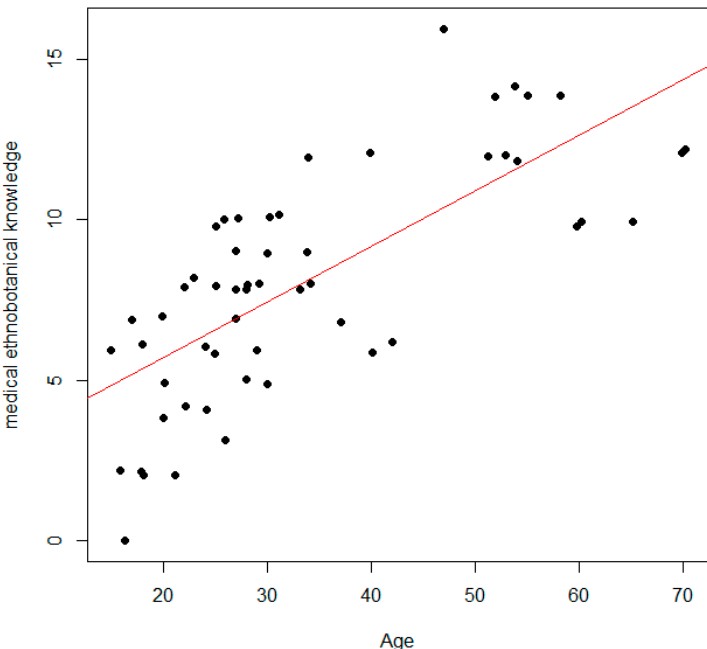

**Figure 2.** Relationship between medicinal ethnobotanical knowledge and age in five selected Waorani communities.

It is interesting to see that the lesser medicinal plant knowledge among young people is a phenomenon that even extends to medical specialist community members, such as is the case with traditional healers in southwestern Ethiopia [8]. As in the prior presented cases, the author gathered that a reduced interest to inherit and use ethno-medicinal knowledge in younger generations might cause this discrepancy in knowledge between older and younger healers.

### 3.5. The Influence of Schooling

Although young people's diminished interest in medicinal plants could be a factor for the demographic differences in knowledge among the Waorani society, formal schooling might be having a greater influence. Results show that the amount of years in school had a significant negative correlation with ethnobotanical knowledge about medicinal plants ($R^2 = 0.10$, $p < 0.01$) (Figure 3).

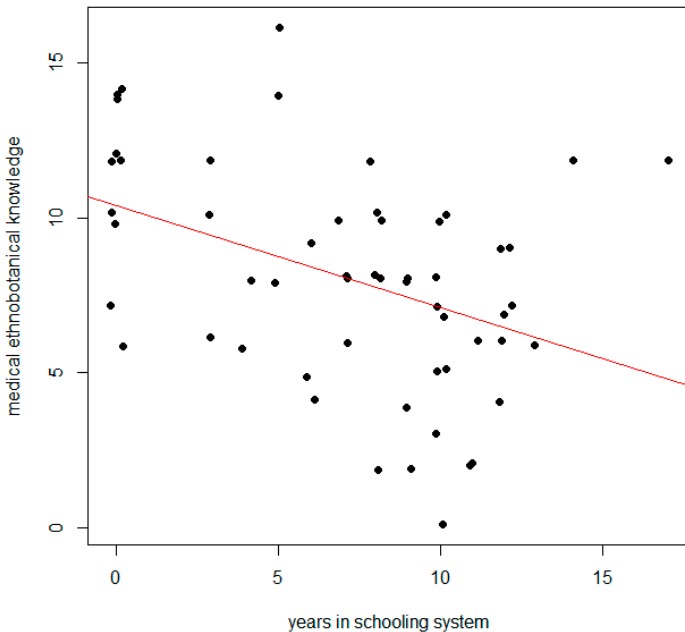

**Figure 3.** Relationship between medicinal ethnobotanical knowledge and years in the schooling system in five selected Waorani communities.

To understand this finding it is necessary to understand the dynamic of transmission for traditional ecological knowledge. In Waorani society, teaching TEK to the younger generations was and is the role of elders, principally parents and grandparents. During the study's fieldwork, it was observed that such knowledge was transmitted during long walks in the forest, for example during hunting or gathering activities.

Other methods, such as observation and imitation, are fundamental principles of the Waorani TEK learning process. Saynes-Vásquez et al. [20] highlight that among the Isthmus Zapotecs in Mexico, children who attend school forgo the potential to learn about the local flora through outdoor activities. He agrees with other authors [21] that time spent in the classroom reduces the opportunities for learning traditional knowledge in the way it has been learned in the past.

Further, the article states that formal educational programs marginalize local knowledge by encouraging a more urban lifestyle that leads to a disinterest in the local natural environment and related knowledge.

In the Waorani communities included in this study, the concept mentioned above does not apply entirely. Even if formal education with a nationally organized school system exists, some keen Waorani teachers promote the learning of traditional ecological knowledge in their language, Wao-tededo. Since this implementation is recent, the field trips and cultural classes cannot yet balance the absence of the traditional transfer of knowledge. As it has been observed in other cases [42], positive effects might be possible to perceive in the future, but until now, in terms of TEK learning, the formal schooling system has been removing children from their traditional cultural context instead of supporting it.

### 3.6. Community Factors Affecting Knowledge

Comparing the two different kinds of communities, more integrated and more isolated, results show that medicinal plant knowledge varies significantly between the communities with a road and easy access to a health center and the communities inaccessible by road and unable to reach a health center in a short amount of time (t = 2.446, df = 52.087; $p \leq 0.05$). The communities inaccessible by road and unable to reach a health center in a short amount of time have a significantly higher medicinal plant knowledge with an average score of 9.03, compared with the average score of 6.8 in the other communities. There are two potential explanations for these findings. First, lower knowledge in villages with road and health center access might be due to lack of use since these people can easily obtain medicines. Studies done by Kinman [43] and Stock [44] about the use of primary health care in Bolivian suburban societies and utilization of health facilities in rural Nigeria, respectively, showed that people living close to health care centers tend to visit them more often and thus neglect the use of medicinal plants. Vandebroek et al. [28] come to the same conclusion in a study about indigenous communities in the Bolivian Amazon. With increasing distances to health care centers, the use of western medicine was less likely while medicinal plant knowledge increased. He concluded that people who were more isolated depended more on traditional rather than western medicine. In any case, the distance was not the only factor. Second, ethnobotanical knowledge can be maintained in villages far from health centers because they do not have transportation facilities nor economic resources to access medical assistance. Vandebroek et al. [28] further explain that one of the reasons for this fact was the lack of opportunities to sell agricultural products, due to low market integration. This makes pharmaceutical products less affordable for inhabitants of more isolated areas. Therefore, the use of medicinal plants for primary healthcare plays a more important role in these areas. These arguments might apply partly to the people living in Quehueri'ono and Apaika. The absence of a health care center in these communities means that people who want to visit a public health care center have no choice but to take long boat and bus rides and pay the correspondent traveling costs. Consequently, for community members, it might be preferable to use traditional medicine instead of shouldering the physical and economic strains linked with the travel to a primary health care center. On the other hand, inhabitants of the communities in the Yasuní National Park can use free buses and reach free primary health care in less than half an hour. It is likely that this convenience encourages them to visit the doctor and use modern medicine instead of spending their time searching for the appropriate plant and preparing the medicine, as some community members stated in the interviews. When traditional medicine is neglected, the practice falls into oblivion. Once again, this absence affects the dynamics of the traditional knowledge transmission process because children do not have the chance to observe which plants have to be gathered and how they have to be prepared.

In addition, the arrival of ecotourism to the communities of Quehueri'ono and Apaika has to be considered through the lens of conservation and loss of ethnobotanical knowledge. In the studied communities inside Yasuní National Park, ecotourism is not present due to the restricted access imposed by the oil company. Some interviewees from Quehueri'ono and Apaika stated that it is important to maintain traditional knowledge and habits because this attracts tourist to visit their community. Due to the communities' physical isolation, it takes more effort to sell agricultural or handicraft products in market towns. Therefore, tourism provides an important source of economic income for the society. With regard to such trends, Voeks & Leony [9] see a disengagement of plant knowledge from its cultural context and have doubts about the success of this marked-based motivation for the conservation of traditional plant knowledge. Even though it might not conform to an outsider's romantic vision that the conservation of traditional knowledge and habits within indigenous communities could rely on economic motivation, the use of their cultural intellectual goods is a legitimate way for the groups to adapt to changes in an increasingly globalized society. Therefore, it is not beyond the bounds of possibility that in some cases, ecotourism could act as an element in the conservation of traditional ecological knowledge.

*3.7. Globalization and Acculturation Process*

The variances in ethnobotanical knowledge between age, between different communities, and the negative correlation of formal education can be explained as a result of earlier and current socio-economic and cultural changes affecting the Waorani society. Missionaries started this process by introducing western values while suppressing traditional ones. The activity of the oil companies brought infrastructure and employment inside the Waorani territory. Waorani working with the oil company personnel have been confronted with new realities and have developed an interest in the materialistic and ideological novelties, which are now finding their way into Waorani culture. Roads built by the companies broke down barriers of physical isolation and enabled traveling and trading options that used to be unimaginable. Therefore, as the Waorani entered the globalized world, they were simultaneously captivated by all of its materialistic facets. The unavoidable adaptions to the new reality caused drastic socio-economic and cultural changes, as well as in the knowledge of ethnobotanical plants. Traditional knowledge was abandoned as it no longer seemed to fit with current realities [16]. For example, medicinal plant knowledge used to be a survival skill, but health problems can now be solved with modern medicine. Additionally, formal education programs follow a nationally mandated curriculum that does not emphasize traditional ecological knowledge as part of the training that prepares students for the realities of a globalized world [9].

## 4. Conclusions

The factors influencing medicinal plant knowledge among the Waorani society include both demographic and socio-economic traits. Older people tend to have a greater knowledge about medicinal plants than their younger counterparts. In addition, formal education, community location, and infrastructure correlate with the ethnobotanical knowledge of the Waorani, whereas the potential interference of ecotourism with the conservation of traditional knowledge requires further investigation. Above all, medicinal plant knowledge is very vulnerable and is in danger of more erosion in the future.

Not every Waorani community is subject to the same physical and socio-economic circumstances, which results in differences in medicinal plant knowledge. Communities that are more physically isolated, inaccessible by road, and without easy access to primary health care, have a significantly higher knowledge than those communities with a road and easy, free access to a Western doctor. It also seems that cultural change affects individuals differently, depending on the demographic profile of the person.

Hence, elders have greater medicinal plant knowledge than the younger generations. This shows an erosion of traditional knowledge and reveals the vulnerability of this exceptional cultural good. On the other hand, it indicates that this knowledge is still part of the Waorani society, which means it has the potential to be conserved. With prompt and appropriate policies, the loss of this valuable knowledge might be minimized or even recovered. Policymakers are advised to take action quickly before this unique set of knowledge fades away.

**Author Contributions:** Conceptualization, H.W., M.B., R.M., and C.B.; methodology, H.W. and C.B.; field work, H.W. formal analysis, H.W.; writing—review and editing, H.W. and C.B.

**Funding:** This research received no external funding.

**Acknowledgments:** I am very thankful to the members of the Waorani communities who participated in the study, especially Amo Enomenga who guided me and translated between Spanish and Wao-tededo. I would also like to give a special thanks to Miguel Ángel Rodriguez Villacreses and the staff of the Estación Científica Yasuní de la Pontificia Universidad Católica de Ecuador for their help with logistics and plant advice, to Margaret Metz and David Robson for advice on statistics, to Carlos Cerón and Manuel Macía for their help in the choice of species, and to Elizabeth Tokarz and Soraia Branco by their support. Isabel Ruiz-Mallen made very useful comments and corrections of several versions of this text. Stephen Bell revised the English and Filipe Carvalho helped us with the cartography.

**Conflicts of Interest:** The authors declare no conflicts of interest.

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
