# Peer review of "Factors Affecting Traditional Medicinal Plant Knowledge of the Waorani, Ecuador"

_sustainability, doi:10.3390/su11164460_

Round 1
Reviewer 1 Report
Review — Waorani medicinal plants
This is a welcome paper about an important topic. Medicinal plants are important for sustainable development of rural areas with important indigenous populations, and understanding factors that drive traditional knowledge about them may contribute to a balanced use of the natural resources in such areas. The paper is well conceived and the research behind it is mostly well executed although the documentation of plant names is conspicuously inadequate. In addition, several points, mostly related to the presentation, need to be addressed.
General comments:
The English writing appears mostly grammatically correct and with few spelling errors. But it is still not “good English”. Many words do not cover exactly what is intended (as far as one can see from the context), and often the wording is imprecise. There are many un-necessary words, more similar to Spanish writing style, and not following English scientific writing style. And often the statements are presented as logical sequences, but the content of them does not reflect a logical reasoning. I have made detailed comments in this respect to the Abstract (see below), but the whole manuscript should be checked for such discrepancies. The text should be corrected by someone who is a native English speaker and, if possible, who understands the subject matter of the paper.
Specific comments:
TITLE:
The paper is not so much about changes and it is about traditional knowledge. I know some use the term “society” for the whole group of Waorani, but I think it is not really an appropriate term. I suggest changing the title to:
Factors Affecting Traditional Medicinal Plant Knowledge of the Waorani, Ecuador
ABSTRACT:
• The abstract is mostly informative – as it should be – but in one place it is indicative – which it should not be. That is in lines 16-17 where it says “… differences among isolated and easily accessible communities.” This should be changed to be informative, i.e. it should mention what were the differences found between the two kinds of communities.
• First sentence (lines 10-11). No need for “the” before “medicinal plant knowledge”. No need for “indigenous society”. No need for “in accordance”. No need for “both”.
• Second sentence (line 12). No need for “both”
• Third sentence (lines 12-14). The numbers “(men n=30, women n=29)” do not belong with the “interviews” but instead it belongs with “people” so it should be moved to after that term. And those who were interviewed were not just people but they were “informants” so the “people” should be changed to “informants”. The “Yasuni Biosphere Reserve” is not shown on Figure 1. One has to read the running text to find out what that is – it ought to be obvious from the figure.
• Fourth sentence (lines15-17). Delete “Results show” and be more direct and say “We found”. Change “an individual’s” to “informants’”. In line 16 insert “informants’” between “between” and “medicinal plants”. Change ”the” to “their”.
• Fifth sentence (lines 17-18). This sentence does not reflect the discussion of the results further on, where the knowledge/age relationship is presented as a general feature and not as something caused by neither socio-cultural changes nor globalization processes.
• Sixth sentence (lines 18-20). The first two relationships are indeed described in the paper, but the data do not include anything about how knowledge is transmitted, so it is not justified to mention that.
• Seventh sentence (lines 21-22). It seems useless to tell policy makers to “take action”, without being specific about what those actions should be. So please make concrete proposals for what needs to be done. As it stands, this sentence is better left out.
KEY WORDS:
Key words help future readers locate the paper through electronic search machines. But these machines also search all words in the titles of papers, so it is a waste to give keywords that repeat any of the words from the title. Here, Waorani and medicinal plants are found in both the title and the keywords – they should be deleted from the key words.
INTRODUCTION:
• Line 31: Voeks and Leony is number [9] in the references and not [10] as indicated here. This pushes all the subsequent numbering to be incorrect and difficult to check.
• Line 34: Change “However” to “Consequently”
• Line 37: Is written here, the sentence suggests that women know more about medicinal plants than they know about men. To say so, is obviously not the intention of the authors. Sentence should be changed to “…, women know more than men know about medicinal plants…”
• Line 38: Lack of logic here. The fact that men are at risk of loosing knowledge does not mean that they have lost knowledge. And what exactly is suggested by this statement – it is unclear.
• Line 39: More common to refer to “homegardens” when talking about ethnic communities.
• Line 80: Ethnobotany is a research field so – even if commonly used – “ethnobotanical knowledge” does not really make sense. Better to use “traditional knowledge” as is also done elsewhere in this manuscript (line 46).
• Line 86: On Figure 1, Yasuni reaches Napo river for a section
Line 86-87: “river Curraray” is not shown of Figure 1; Change South to south.
• Line 90: “mainstream society” may not be the best choice of words.
Line 92: “Waorani territory” – what is that? Not shown on Figure 1.
MATERIALS AND METHODS:
Lines 106-107: Figure 1 is mostly OK, but – as a general rule – the best figures to illustrate scientific papers are such ones that include all geographic localities named in the text and that does not include any names of places not mentioned in the text. Figure 1 can be improved in this respect, For Instance the Waorani Biosphere Reserve is mentioned in the text but not on the map, river Curraray is in the text but not on the map. Maxus road is on the map but not in the text.
• Lines 124-125: What is “n” – I suppose informants but then write that; “street” is called “road” in the text; travel time repeated 3x – better change “Distance“ to “Travel time to”
• Line 134: Why not name the academic experts, and by the way - were their contribution not important enough for them to merit co-authorship?
• Line 141 (Table 2): Errors in plant names should be corrected. Astrocarium should be Astrocaryum; Curare should be Curarea; Eucharis grandiflora should be Eucharis x grandiflora; albivensis should be albivenis; subincarum should be subincanum.
• Line 141 (Table 2): Growth form: Bush should be Shrub; Herbaceous plant should be Herb;
• Line 141 (Table 2): cultivate plant should be cultivated
• Line 141 (Table 2): Use: cavaties should be cavities
• Line 141 (Table 2): The plant names appear without authors names which is customary in scientific writing.
• Line 140 (Table 2): There are no voucher specimens which is also customary in ethnobotanical scientific writing (“without vouchers, plant names are fiction”). Most journals do not accept papers that do not document the names for the used plants with voucher specimens and indication of which herbarium they were deposited in. How were the plants identified in the field? How can we be sure that the information really related to the species mentioned in Table 2?
• Line 143: Change “interviewee” to “informant” throughout. Also, in Line 148 where they are called “representatives”
• Line 148-152: This information is largely repeated from Table 1. It is a golden rule that Tables, Figures and Text should not overlap in the information they provide.
• Line 159: How were these pictures produced? How did you secure that the pictures really showed the species they were supposed to show? Here voucher specimen documentation is essential to secure that the right names were used for the right plants.
• Line 174: Be specific and say that the Wao name of the plants were not considered.
RESULTS AND DISCUSSION:
• Line 188: Plant name spelled incorrectly.
• Line 190: Not appropriate to talk about “correct” answers, when what you mean (I suppose) is that the answers. Agreed with what had previously been reported in the literature (the eight references consulted).
REFERENCES:
The numbers for the references cited in the text do not agree with numbers given in the list of references.
• Line 429: Volume and pages missing
Author Response
Thank you very much for the revision, we appreciate to much your comments and suggestions. We have done several changes in order to incorporate the comments and suggestions of the two referees who have reviewed our manuscript. Below we have indicated the changes done since your comments.
1. About the English. Now a native English speaker has revised whole manuscript.
2. Title: we have change it with your proposal. Thanks.
3. Abstract: we have try to get a more informative abstract. We have done all the changes that you has proposed about words and concepts. Lines 17-22, has been deleted as you suggest. Thanks.
4. Key words: we have delete the words you has proposed. Thanks.
5. Introduction: we have introduced your changes proposals. About Figure 1, the map of Yasuni, it is from Santiago Espinoza (as we have indicated) and we do not have any possibility to change it. We have change “mainstream” by contemporary. We have deleted “Waorani territory”. Many thanks for your comments.
6. Material and methods: all your suggestions either in the text as in table 2, has been applied. The academic experts who help us to the plant selection were Carlos CerĂłn and Manuel MacĂa, both are in acknowledgements. The question about co-authorship of academics is a difficult issue. We will consider these aspect in next releases.
Regarding voucher specimens, the 10 selected plants were easy to identify; anyway, during the fieldwork, the researcher at charge, HW, was located at Yasuni Scientific Station (PUCE University) where there are several botanic experts working all along the year which and we have its support. The pictures of the plants were corroborated by the staff of the Scientific Station.
We have deleted repeated information; also we have indicated that Wao name were no considered. Thanks for the comments and corrections. We hope to have answered correctly.
7. Results and Discussion: we have spelled correctly the plant name. We have change the mistake regarding “correct” answers. Thanks.
8. References: Now all the references are correctly numerated and cited. Many thanks.

Reviewer 2 Report
The authors explored the shift in TEK related to medicinal plants in select Waorani communities in Ecuador. This is a global phenomenon and could be applied to nearly all indigenous populations in any country or region. The paper highlights some important considerations related to globalization and access to modern conveniences that directly influence their TEK for medicinal plants.
I felt that the paper was very well written and I had only minor edits indicated on the PDF itself.

Author Response
Thank you very much for the revision, we appreciate to much your comments and suggestions. We have done several changes since yours comments, especially those relatives to the statistic that we have reformatted entirely due to your suggestions about some mistakes.
1. About Figure 1, the map of Yasuni, it is from Santiago Espinoza (as we have indicated) and we do not have any possibility to develop many changes.
2. We have enlarged the caption of table 1 in order to be more concise. We have spelling correctly the name of community, in the table and text, as you suggest. Thanks for the consideration.
3. Effectively, finally we analyze the knowledge of 10 plants. We have change table 2.
4. In the last attached version of the manuscript we have solve formatting problems.
5. Regarding at your question on Waorani fathers, it is just for statistical’s, in order to see the effect of this. Finally there were no significant differences if both parents were or not Waorani.
6. We have included the value of statistics in the test, not just the p-value.
7. The repeated sentence relative at the work of Hanazaki was deleted from the “gender and knowledge” block Results and Discussion. Thanks for the observation.
8. About the number of points in figure 2 and 3, effectively there are some overlapped points.
9. Regarding at the values of the relationship between cross-cultural marriages and knowledge we have undergo the statistics another time and now it’s ok. Effectively, we did some mistakes there.
10. At caption of figure 2, we have clarified the fact that we are using only five Waorani communities.
11. We have rewritten the sentence of line 267.
12. Caption of figure 3 has been enlarged.
13. As you suggested, minor mistakes and not precise words on the text, has been changed.
14. We have improve the reference of Coe and Anderson (1996).

Round 2
Reviewer 1 Report
Review — R1-Waorani medicinal plants
General comments:
The English writing has been improved in this revised (R1) version of the manuscript but there are still several places where it could be further “massaged” to become “good English”. There are still words that do not cover exactly what is intended (e.g. line 41 “individual schooling” may occur in traditional contexts – so I assume what you mean is “westernized schooling”?; line 53 . “inside” should be “in”; line 53 “about” should b “of”; Line 57 “within the” should be “what is now”; Line 58 change “They have been” to “There they have become”). There are still many un-necessary words, - I have stricken some of them with a red line. I think the authors should. Ask their helper, who is a native English speaker to read it once more.
ABSTRACT:
• Line 12:Delete “Fifty-nine”
• Line 13: Change “people” to “56 informants”
• Line 14: Add “Waorani Ethnic Reserve”
• Line 16: Change “seen in” to “thought to be”
MATERIALS AND METHODS:
•Figure 1: A good map illustratoin an article should show the location of all geographic names mentioned in the text, AND it should not have anything on it which is not in the text. It is no excuse that the map is derived from someone else’s work. The correct things to do is to make a new map which fulfils the criteria mentioned her. And the names on the map and in the text should coincide ( not as in Via Maxus/Maxus road)
•Table 2): “cultivate” for Musa should be “cultivated””
• I am still not happy that voucher specimens were not collected. But as you explain “pictures” were shown to the informants, and the names of the plants shown on the pictures were verified by botanical experts at the Yasunic Scientific Station. This should be mentioned in the text.
• Line 159: How were these pictures produced? How did you secure that the pictures really showed the species they were supposed to show? Here voucher specimen documentation is essential to secure that the right names were used for the right plants.
• Line 169: Change “inteviewees” to “informants”
Author Response
Dear referee, thank you very much by this second in deep revision. We are going to answer each of your considerations.
General comments:
The English writing has been improved in this revised (R1) version of the manuscript but there are still several places where it could be further “massaged” to become “good English”. There are still words that do not cover exactly what is intended (e.g. line 41 “individual schooling” may occur in traditional contexts – so I assume what you mean is “westernized schooling”?; line 53 . “inside” should be “in”; line 53 “about” should b “of”; Line 57 “within the” should be “what is now”; Line 58 change “They have been” to “There they have become”). There are still many un-necessary words, - I have stricken some of them with a red line. I think the authors should. Ask their helper, who is a native English speaker to read it once more.
Yes, finally the manuscript has been carefully revised by a Englsih native in order to become a "good English". Additionally we have change also your suggestions. We hope that in the new version all the text is fine.
ABSTRACT:
• Line 12:Delete “Fifty-nine”
• Line 13: Change “people” to “56 informants”
• Line 14: Add “Waorani Ethnic Reserve”
• Line 16: Change “seen in” to “thought to be”
Done. Thanks.
MATERIALS AND METHODS:
•Figure 1: A good map illustratoin an article should show the location of all geographic names mentioned in the text, AND it should not have anything on it which is not in the text. It is no excuse that the map is derived from someone else’s work. The correct things to do is to make a new map which fulfils the criteria mentioned her. And the names on the map and in the text should coincide ( not as in Via Maxus/Maxus road)
Finally we have done a new map where all the localities and correct names of the toponymy is
arranged.
•Table 2): “cultivate” for Musa should be “cultivated””
Done
I am still not happy that voucher specimens were not collected. But as you explain “pictures” were shown to the informants, and the names of the plants shown on the pictures were verified by botanical experts at the Yasunic Scientific Station. This should be mentioned in the text.
Ok, we have add this at the text.
Line 159: How were these pictures produced? How did you secure that the pictures really showed the species they were supposed to show? Here voucher specimen documentation is essential to secure that the right names were used for the right plants.
During the field work we were at YasunĂ Scientfic Station where there are several botanical experts. They were asked for the identification and for the wao names in order to have more information during the intervews . Staff of Scientific Station have a good level of knowledge of plants so we trust in them. In next researchs we are going to consider this interesting approach.
Line 169: Change “inteviewees” to “informants”
Done.
